# A tuberculin skin test survey among healthcare workers in two public tertiary care hospitals in Bangladesh

Md Saiful Islam[1,2¤]*, Abrar Ahmad Chughtai[2], Arifa Nazneen[1], Kamal Ibne Amin Chowdhury[1], Muhammad Tauhidul Islam[1], Sayeeda Tarannum[1], S. M. Hasibul Islam[1], Sayera Banu[1], Holly Seale[2]

1 Infectious Diseases Division, Program for Emerging Infections, icddr,b, Dhaka, Bangladesh, 2 School of Population Health, University of New South Wales, Sydney, Australia

¤ Current address: School of Population Health, University of New South Wales, Sydney, Australia
* saiful@icddrb.org

**Data Availability Statement:** A minimal dataset that supports the study has been attached as a Supporting Information file. icddr,b's department

## Abstract

In Bangladesh, there is currently no data on the burden of latent TB infection (LTBI) amongst hospital healthcare workers (HCWs). This study aimed to determine the prevalence of LTBI and compare the prevalence among HCWs in two public tertiary care hospitals. Between September 2018 and August 2019, we conducted a cross-sectional study in two public tertiary care general hospitals. Using a survey and tuberculin skin test (TST), we assessed risk factors for LTBI, adjusting for known and plausible confounders. In addition, a facility assessment was undertaken to understand the implementation of relevant IPC measures. The prevalence of LTBI among HCWs was 42%. HCWs spent a median of 6 hours (SD = 1.76, IQR 2.00) per day and attended an average of 1.87 pulmonary TB patients per week. HCWs did not receive any TB IPC training, the wards lacked a symptom checklist to screen patients for TB, and no masks were available for coughing patients. Seventy-seven percent reportedly did not use any facial protection (masks or respirators) while caring for patients. In the multivariable model adjusting for hospital level clustering effect, TST positivity was significantly higher among HCWs aged 35–45 years (aOR1.36, 95% CI: 1.06–1.73) and with >3 years of service (aOR 1.67, 95% CI: 1.62–1.72). HCWs working in the medicine ward had 3.65 (95% CI: 2.20–6.05) times, and HCWs in the gynecology and obstetrics ward had 2.46 (95% CI: 1.42–4.27) times higher odds of TST positivity compared to HCWs working in administrative areas. This study identified high prevalence of LTBI among HCWs. This may be due to the level of exposure to pulmonary TB patients, and/or limited use of personal protective equipment along with poor implementation of TB IPC in the hospitals. Considering the high prevalence of LTBI, we recommend the national TB program consider providing preventative therapy to the HCWs as the high-risk group, and implement TB IPC in the hospitals.

of research administration maintains a data repository and a copy of the complete dataset will remain in the repository. Interested researchers may contact Ms. Armana Ahmed, head of research administration (aahmed@icddrb.org), for approval and full data access.

**Funding:** This research protocol was funded by the United States Centers for Disease Control and Prevention (CDC), through the cooperative agreement grant number 5U01GH1207. icddr,b acknowledges with gratitude the commitment of CDC to its research efforts. icddr,b is also grateful to the Governments of Bangladesh, Canada, Sweden, and the UK for providing core/unrestricted support. The funders had no role in study design, data collection and analysis, decision to publish, or preparation of the manuscript.

**Competing interests:** There is no conflict of interest among the authors.

## Introduction

In 2019, the WHO estimated that 10 million people developed TB disease globally, of whom 44% were from South-East Asia [1]. Moreover, 22,314 healthcare workers (HCWs) developed TB in the same year, with most coming from high TB burden countries in Asia [1]. Hospital HCWs in high TB burden countries are at increased risk of TB infection due to their exposure to a higher number of pulmonary TB patients than the hospital HCWs working in low TB-incidence countries [2]. A recent mathematical modelling study on the global burden of latent TB infection (LTBI) estimated that around 1.7 billion people are infected with LTBI in the world [3]. In a systematic review of 18 studies from seven high TB burden countries, the prevalence of LTBI among HCWs was reported to be 47% (95% CI 34–60) [4]. This risk may be high among HCWs who work in health facilities that lack proper infrastructure and limited implementation of TB infection prevention and control (IPC) healthcare measures. The risk of LTBI among different health care workers may vary by place of work, duration of exposures, and compliance with TB IPC measures [5]. Prior studies identified considerable heterogeneity in the risk of LTBI among different occupations and reported high risk among doctors, nurses, and ancillary workers [5, 6]. HCWs that were most likely to be infected had the most prolonged duration and extent of patient contact [6, 7].

People with LTBI represent a reservoir for potential TB disease [3]. Without treatment, about 5 to 10% of infected persons may develop TB disease at some point in their lives, and the active stage frequently occurs within the first two years after infection [8, 9]. People with an impaired immune system are at increased risk of developing TB disease than persons with standard immune systems [9]. Diabetes and smoking also increase TB disease risk among the person with LTBI [10].

There is no gold-standard test to diagnose LTBI [11]. The widely available diagnostic tools are tuberculin skin test (TST) and interferon-gamma release assay (IGRA) that measure the response to *in vivo* or *in vitro* stimulation by *M. tuberculosis* antigens [12–14]. IGRA has some advantages over TST. IGRA does not require return visits and is not affected by Bacille Calmette-Guérin (BCG) vaccination status or infection with non-tuberculous *Mycobacteria* [13, 14]. However, IGRA is costly and needs a specialized laboratory for sample processing that is not widely available. Therefore, the American Thoracic Society (ATS), Infectious Diseases Society of America (IDSA), and the US Centers for Disease Control and Prevention (CDC) clinical guidelines recommend TST when an IGRA is not cost-effective or the laboratory supports required for IGRA are unavailable [15]. TST has widely been used to screen HCWs for LTBI in low- and middle-income high TB incidence countries [16, 17].

Although known pulmonary TB patients usually are not managed at public tertiary care hospital wards, a substantial number of pulmonary TB patients are admitted to these hospital wards before TB diagnosis and for other TB complications. Moreover, in health care facilities, a substantial number of pulmonary TB patients may remain undiagnosed due to a low index of suspicion and a lack of proper diagnostic evaluation. For example, in a study conducted in Thailand, 73% of patients hospitalized with clinical pneumonia were not adequately evaluated for TB, and an estimated 2%–12% of them had smear-positive, pulmonary TB [18].

Bangladesh shares 3.6% of the global total of 10 million people estimated with TB diseases in 2019 [1]. Bangladesh is one of the 22 high TB burden countries in the world with an estimated incidence for all forms of TB in 2019 was 221 (uncertainty interval: 161–291) per 100 000 population [1]. In line with the country's TB epidemiology, Bangladesh's extended program on immunization has included neonatal bacillus Calmette-Guérin (BCG) vaccination nationwide since the 1980s [19]. The national coverage of the BCG vaccine was 86% in 1991; 95% in 2000; and 99% since 2013 [19].

Public tertiary care hospitals in Bangladesh are overcrowded; lack necessary airborne IPC measures and hand washing stations; therefore, HCWs working in these hospitals are at increased risk of LTBI infection [20]. The approximate risk for persons with LTBI to develop active TB during their lifetime is 10% [21], and therefore, LTBI is considered the starting point for active TB disease [22]. Identifying HCWs workers with LTBI is an essential component of TB IPC in health settings [23]. However, there is no data on the burden of LTBI amongst HCWs from public tertiary care hospitals in Bangladesh. Therefore, we aimed to determine the prevalence of LTBI and compare the prevalence among different groups of HCWs in two public tertiary care hospitals in Bangladesh.

## Materials and methods

### Study design and sites

Between September 2018 and August 2019, a cross-sectional study was conducted in two 1000-bed public tertiary care hospitals (Hospital A and Hospital B) in Rajshahi and Barisal in Bangladesh. We selected these hospitals due to the following reasons: (1) these sites admit around 400 TB patients each year as inpatients, the majority of which are pulmonary; (2) there is already disease surveillance being conducted, including for MDR-TB; and (3) they are the largest tertiary care referral hospitals at the division levels and provide care to patients including those with TB.

### Study definitions, participant recruitment, and screening

Any person aged ≥17 who delivered care and services to patients in the hospital and received payment was defined as a HCW. We included doctors, nurses, interns, ancillary workers (cleaners, general non-medical ward staff, female attendants [known as ayas]), laboratory staff, and administrative workers. We organized meetings with the HCWs in each of the hospitals to share the proposed study details and facilitate recruitment.

### Sample size

Based on a recently completed study on the prevalence of LTBI of 54% among HCWs in TB specialty hospitals, we calculated the sample size needed for this study, allowing for a 95% confidence level and 5% absolute precision [24]. To estimate the prevalence within five percentage points of the hypothesized prevalence along with a 20% refusal, and the hospital level cluster effect as 1.5, the required sample size was 688.

### Data collection

A questionnaire was used to collect data on demographics, history of BCG vaccination, duration of service, history of living with a TB positive person at home, and exposures to TB patients in the community. The tool, adopted from a previous study, was updated and pretested for cognitive test among five HCWs in the study hospitals [25]. In addition, participants were asked to complete a diary card for 7 days, to record their exposures to TB patients and IPC practices after the TST. To supplement the quantitative design, we conducted a TB IPC facility assessment using the WHO recommended checklist for periodic TB IPC assessment in health settings [26]. We assessed the implementation of structural, administrative, and environmental control measures through in-depth interviews with a range of stakeholders within the hospital (i.e., department heads, nursing heads, etc.), and direct observation of TB IPC practices. Finally, the team also mapped inpatient wards to describe the physical layout, including the distribution of patients' beds, doors, windows, and HCWs workstations.

## TST Procedure

TST that uses purified protein derivative to elicit immune reactions is less expensive, easy to perform, and widely used for TB screening in low-and middle-income countries, including Bangladesh [27]. We used the TST for detecting LTBI according to the WHO and the International Union against Tuberculosis and Lung Disease recommendations for TST surveys in high TB burden countries [28]. We used the Mantoux technique and administered a ten-tuberculin unit dose (0.1 ml) of RT23 Purified Protein Derivatives (Arkray Healthcare Pvt. Ltd., Sachin, Gujarat, India) for study participants. The team organized a meeting in the study hospitals and informed the HCWs about our study objectives and planned activities. The team visited the hospital director's office, collected a list of HCWs, and then invited the HCWs by visiting the wards. After seeking informed written consent, a trained medical technologist administered the TST into the forearm. At 48 to 78 hours following TST placement, the technologists called each of the HCWs to know their present duty station. The team then visited the workplace and measured HCWs' transverse diameter of induration using a caliper [29]. We defined a positive TST as induration ≥10 mm as recommended by ATS, IDSA, and CDC [30, 31]. We dichotomized TST results as positive or negative.

## Data entry and analysis

The research team entered the data into Microsoft Excel. We used descriptive statistics to summarize the distribution of demographic and clinical variables, along with variables measuring exposure to TB. We estimated odds ratios (ORs) with 95% confidence intervals (CIs) as a measure of association between factors (characteristics, duration of service, occupational TB exposure, community TB exposure) using univariate logistic regression. In the multivariate analysis, we used a generalized linear model adjusting for hospital level clustering effect (using 'glm.cluster' command from the *miceadds* R package). We included factors with an association with a p-value <0.20 in the unadjusted analyses with TST positivity along with LTBI risk factors identified in the prior published literature [32]. To check multicollinearity, we performed a variance inflation factor (VIF) analysis, and factors with VIF> 5, we considered them collinear and excluded from the multivariate model [33]. The open-source statistical package R version 3.6.3 (R Foundation for statistical computing, Vienna, Austria, Available at https://www.Rproject.org/) was used for analysis.

## Ethics

We sought informed written consent before the TST and participation in the questionnaire or diary cards survey. The institutional review board of the International Centre for Diarrheal Diseases Research, Bangladesh (icddr,b), and the University of New South Wales ethical review committee approved the study (PR-16090 and HC # HC180517).

## Results

Of the 4,057 HCWs listed in the hospital register, we approached 977 (24%) for the TST survey. Among the 977 HCWs, 84% (818/977) consented and participated in the TST survey. One hundred and fifty nine HCWs did not consent for TST due to their illness, prior history of allergic reaction to TST, unavailability for TST reading, and past history of active TB disease. The participants' median age was 28.8 years (standard deviation, SD = 10.87 years; range: 17 to 62 years), and 73% were female. The largest group was nurses (65%) followed by ancillary workers (18%), doctors (13%), and laboratory staff (4%). Eighty-one percent reported receiving the BCG vaccine as a child (Table 1). Seven percent of the participants had history of

**Table 1. Baseline characteristics of HCWs at two public tertiary care general hospitals in Bangladesh, 2019.**

| Demography and exposures | TST participants (N = 818) % (n/N) |
|---|---|
| **Location of facilities** | |
| Hospital A | 50 (409/818) |
| Hospital B | 50 (409/818) |
| **Sex** | |
| Male | 27 (218/818) |
| Female | 73 (600/818) |
| **Education** | |
| 0 to Primary | 3 (24/816) |
| Secondary | 12 (98/816) |
| HSC/Diploma | 59 (485/816) |
| MBBS/BA/BSC/Hons and above | 26 (209/816) |
| **Occupational group** | |
| Doctors | 13 (104/818) |
| Nurse | 65 (534/818) |
| Ancillary workers | 18 (150/818) |
| Laboratory staff | 4 (30/818) |
| **Age in years** | |
| Median, IQR | 28.8 (13.8) |
| >25 | 30 (245/818) |
| 25–35 | 39 (321/818) |
| 35–45 | 14 (112/818) |
| ≥45 | 17 (140/818) |
| **Years working as HCWS** | |
| ≤ = 1.5 | 27 (224/818) |
| 1.5–3 | 38 (308/818) |
| ≥ = 3 | 35 (286/818) |
| **Place of work** | |
| Medical ward | 73 (596/814) |
| Gayne and Obstetric | 9 (72/814) |
| Administration | 10 (79/814) |
| Lab | 6 (52/814) |
| ICU | 2 (15/814) |
| **Known exposures to TB patients** | |
| Hospital | 75 (422/566) |
| Community | 03 (18/566) |
| Both hospital and community | 22 (126/566) |
| **History of smoking** | |
| Yes | 7 (60/817) |
| No | 93 (757/817) |

smoking. Active TB was not detected in any of the participants during the study period. Sixty-nine percent (566/818) reported known TB exposure to pulmonary TB patients. Among them, 3% (18/566) had sole exposures to TB patients in the community, and 22% (126/566) had exposures to TB patients both in community and hospital. Those who had sole exposures to known TB patients in the community, 56% (10/18) had TST positivity.

The overall prevalence of LTBI in the two study hospitals was 42%. The median diameter of induration among participants testing positive by TST was 13 mm (IQR 11 to 33 mm). In the

bivariate analysis, the TST positivity significantly differed by age, duration of service, higher income, and medical ward work. Working as ancillary workers or laboratory workers was protective in the bivariate analysis (Table 2). HCWs in Hospital B had a higher TST positivity compared to HCWs in hospital A, although the difference was not statistically significant (odds ratio [OR] 1.19 with 95% confidence interval [CI] 0.89–1.57). Moreover, the history of BCG vaccination, education, known TB exposures to pulmonary TB patients were not statistically significantly associated with TST positivity.

After adjusting for hospital level clustering effect, we noted an increasing trend for TST positivity with increasing age in the multivariate model. Healthcare workers aged 35–45 were statistically significantly associated with positive results (aOR 1.36, 95% CI: 1.06–1.73). A similar trend was also noted with an increasing year of working as HCWs. HCWs with more than three years of service in the hospital were 1.67 times (95% CI: 1.62–1.72) more likely to be positive with TST when compared with HCWs with less than 1.5 years of service. HCWs working in the medicine ward had 3.65 (95% CI: 2.20–6.05) times, and the HCWs working in the gynecology and obstetrics ward had 2.46 (95% CI: 1.42–4.27) times higher odds of TST positivity compared with HCWs working in administrative areas (Table 2). Females were more likely than males (44% Vs. 37%) to have positive results by TST and the difference was statistically significant (OR = 1.08, 95% CI: 1.01–1.18) in the multivariate model. HCWs involved in sputum collection had 1.36 (1.08–1.72) times higher odds of TST positivity compared with HCWs who were not involved in sputum collection.

The facility assessment findings showed that a TB infection control committee did not exist in any facility. None of the HCWs received any training on TB IPC measures. There were no symptom checklists in place to screen patients for TB and no tissues, pieces of cloth, or face masks available for coughing patients. The dairy card findings showed that HCWs spent a median of 6 hours (SD = 1.76, IQR 2.00), of which a median of 3.58 hours was spent in patient contact (SD = 2.50, IQR 2.62) in each day. Each day the HCWs attended a median of 22.5 general patients (SD: 43.02, IQR: 26.75). During a week, the HCWSs provided care to an average of 1.87 pulmonary TB patients in the study wards. We found smear-positive pulmonary TB patients were admitted in the wards, and their median duration of hospital stay was 4.5 days. Seventy-seven percent of the respondents reported that they did not use any medical mask or respirators while caring for patients in the hospital. We also found nurses' duty station inside the patient ward in the study wards that allow them longer exposures to the air space shared by pulmonary TB patients.

## Discussion

The prevalence of LTBI in our study population was higher than what has been previously reported in other high burden TB countries. For example, studies of LTBI among HCWs in public tertiary care hospitals in India reported the prevalence at 20%, and in Pakistan, the prevalence at 40% [34, 35]. The occupational factors associated with TST positivity identified in our study suggest healthcare associated LTBI. The findings warrant immediate implementation of TB IPC in medicine, gynecology, and obstetrics wards. Besides, HCWs positive with TST can be targeted for preventative therapy to prevent active TB progression that may further prevent nosocomial and occupational TB transmission [36].

With the recent evidence of increasing multidrug-resistant TB worldwide, international health agencies, including WHO and the Stop TB Partnership, emphasized implementing TB IPC programs in health settings [37–39]. The WHO 2019 updated TB IPC guidelines, and the guidelines for programmatic management of LTBI recommend preventative treatment for people exposed to *M. tuberculosis* to reduce the burden of TB disease [40, 41]. Worryingly, a

**Table 2. Prevalence of TST positivity and the factors associated with the positivity among HCWs in two tertiary care hospitals, Bangladesh, 2018–2019.**

| Demography and exposures | TST positive % (n/N) | TST negative % (n/N) | OR*(95% CI*) | aOR*(95% CI*) |
|---|---|---|---|---|
|  | 42 (347/818) | 58 (471/818) |  |  |
| **Location of facilities** |  |  |  |  |
| Hospital A | 40 (165/409) | 60 (244/409) | **Reference** |  |
| Hospital B | 45 (182/409) | 52 (227/409) | 1.19 (0.89–1.57) |  |
| **Sex** |  |  |  |  |
| Male | 37 (81/218) | 63(137/218) | Reference |  |
| Female | 44 (266/600) | 56 (334/600) | 1.35 (0.98–1.86) | 1.08 (1.01–1.18) |
| **History of BCG vaccination** |  |  |  |  |
| Yes | 44 (289/659) | 56 (370/659) | 1.38 (0.93–2.08) | 1.17 (0.47–2.94) |
| No | 36 (44/122) | 64 (78/122) | Reference |  |
| Don't know | 38 (14/37) | 62 (23/37) | 1.08 (0.50–2.29) | 0.95 (0.25–3.64) |
| **History of smoking** |  |  |  |  |
| Yes | 35 (21/60) | 65(39/60) | 0.71 (0.40–1.22) |  |
| No | 43 (326/757) | 57 (431/757) | Reference |  |
| **Occupational group** |  |  |  |  |
| Doctors | 39 (40/104) | 61 (64/104) | 0.73 (0.48–1.13) |  |
| Nurse | 46 (245/534) | 54 (289/534) | Reference |  |
| Ancillary workers | 36 (54/150) | 51 (96/150) | **0.66 (0.45–0.96)** |  |
| Laboratory Staff | 27 (8/30) | 73 (22/30) | **0.43 (0.18–0.94)** |  |
| **Education** |  |  |  |  |
| 0 to Primary | 42 (10/24) | 58 (14/24) | 1.06 (0.44–2.49) |  |
| Secondary | 39 (38/98) | 61 (60/98) | 0.94 (0.57–1.58) |  |
| Higher Secondary | 44 (215/485) | 56 (270/485) | 1.18 (0.85–1.65) |  |
| Honours and above | 40 (84/209) | 60 (125/209) | Reference |  |
| **Years working as HCWS** |  |  |  |  |
| 0–1.5 | 38 (85/224) | 62 (139/224) | Reference | Reference |
| 1.5–3 | 38 (116/308) | 62 (192/308) | 0.99 (0.69–1.41) | 0.96 (0.60–1.55) |
| > = 3 | 51 (146/286) | 49 (140/286) | **1.70 (1.20–2.44)** | **1.67 (1.62–1.72)** |
| **Hours working per day** |  |  |  |  |
| <6 hours | 43 (15/35) | 57(20/35) | **Reference** |  |
| 6 hours and above | 43(303/704) | 57 (401/704) | **1.01 (0.51–2.03)** |  |
| **Age in years** |  |  |  |  |
| <25 | 33 (82/245) | 67 (163/245) | Reference |  |
| 25–35 | 40 (128/321) | 60 (193/321) | 1.32 (0.93–1.87) | 1.27 (0.82–1.96) |
| 35–45 | 50 (56/112) | 50 (56/112) | **1.99 (1.26–3.14)** | 1.36 (1.06–1.73) |
| ≥45 | 58 (81/140) | 42 (59/140) | **2.73 (1.78–4.20)** | **2.27 (0.81–6.40)** |
| **Income** |  |  |  |  |
| <10000 | 37 (68/184) | 63 (116/184) | Reference | Reference |
| 10001–20000 | 36 (88/247) | 64 (159/247) | 0.94 (0.63–1.40) | 0.97 (0.68–1.38) |
| 20001–30000 | 45 (110/245) | 55 (135/245) | 1.39 (0.94–2.06) | 1.11 (0.91–1.35) |
| 30001–40000 | 57 (58/101) | 43 (43/101) | **2.30 (1.41–3.79)** | 1.44 (1.00–2.09) |
| 40001 and Above | 56 (23/41) | 44 (18/41) | **2.18 (1.10–4.37)** | 1.89 (1.53–2.34) |
| **Place of work** |  |  |  |  |
| Administration | 34 (27/79) | 66 (52/79) | Reference | Reference |
| Gynae and Obs | 39 (28/72) | 61 (44/72) | 1.22 (0.63–2.39) | **2.46 (1.42–4.27)** |
| ICU | 13 (2/15) | 87 (13/15) | 0.29 (0.04–1.18) | 0.80 (0.36–1.77) |
| Lab | 29 (15/52) | 71 (37/52) | 0.78(0.36–1.65) | 0.67 (0.16–2.75) |

(*Continued*)

**Table 2.** (Continued)

| Demography and exposures | TST positive % (n/N) | TST negative % (n/N) | OR* (95% CI*) | aOR* (95% CI*) |
|---|---|---|---|---|
| Medical Ward | 46 (272/596) | 54 (324/596) | **1.62 (1.01–2.68)** | **3.65 (2.20–6.05)** |
| **Known exposure to TB** | | | | |
| Hospital | 43 (183/422) | 57 (239/422) | Reference | |
| Community | 56 (10/18) | 44 (8/18) | 1.26 (0.85–1.89) | |
| Both hospital and community | 49 (62/126) | 51 (64/126) | 1.63 (0.63–4.35) | |
| **Involved in sputum collection** | | | | |
| Yes | 50 (63/126) | 50 (63/126) | 1.44 (0.98–2.11) | 1.36 (1.08–1.72) |
| No | 41 (283/691) | 59 (408/691) | Reference | |
| **Use of mask or respirator** | | | | |
| Yes | 40 (75/190) | 60 (115/190) | 0.85 (0.61–1.19) | |
| No | 43 (271/626) | 57 (355/626) | Reference | |

person latently infected with multi-drug resistant *M. tuberculosis* may be resistant to some available therapies and warrant optimal TB IPC program to limit nosocomial infection [42].

The study identified that the TST prevalence increases as the number of years working as HCWs increases. Our findings are consistent with prior studies in similar settings where more years of work as healthcare workers were associated with a higher prevalence of LTBI [43, 44]. The lower prevalence of LTBI in HCWs with lesser job duration could be due to lower cumulative exposure to pulmonary TB patients in the hospital. The findings showed public tertiary care hospitals admit pulmonary TB patients, and the patients stay in the hospital for over four days before they are diagnosed with TB, initiate TB treatment, and referred to TB specialty hospitals. The presence of pulmonary TB patients in the medicine wards and the poor implementation of TB IPC measures such as irregular use of N95 respirators and lack of training on TB IPC all are likely to have contributed to this healthcare associated LTBI [45].

Our analysis also identified a statistically significant association between an increase in age and TST positivity, which has also been reported previously in other settings [46, 47]. In the study hospitals, most HCWs joined work when they were young adults (usually between 17 to 21 years). Therefore, the increasing risk of TST positivity with increasing age could be better explained by prolonged cumulative exposure from both occupational and non-occupational exposures [47]. In our study, 22% of the respondents reported known exposures to TB patients both in the hospital and in the community. In Bangladesh, HCWs often hold dual jobs at government health facilities and private practice, including clinics and private chambers. They spend substantial time in private practices that had not been captured in our data [48, 49]. The private clinics and chambers often lack standard TB patient management practice and a limited supply of medical mask and N95 respirators that might have increased the risk of TB transmission [50, 51].

Working in gynecology/obstetrics (OR 2.46, 95% CI 1.14–5.40) wards was another significant predictor for LTBI. Our findings are consistent with prior studies where HCWs from gynecology and obstetrics wards were at increased risk of LTBI [52]. A perception exists that the gynecology and obstetrics ward has a low risk of TB transmission [52, 53]. Therefore, this false sense of security might have de-motivated HCWs following TB IPC measures and put them at increased risk of infection [53]. Also, this study further supports the findings that exposure during procedures such as sputum collection increases the risk of TST positivity among HCWs [54].

Our study suggests nurses could be at higher risk of TB exposures as their working stations were inside the medical wards for administrative work, direct and indirect patient care. The

nurses spend around 6 hours daily for direct and indirect patients care. Though not statistically significant, the higher prevalence of LTBI among nurses could be due to this infrastructural barrier that provided nurses with more exposure to all types of patients, including pulmonary TB patients, than either doctors or other groups of HCWs [20].

This study has several limitations. Firstly, we used a one-step TST in this study. Suppose a person is tested with one-step TST after many years of infection with *Mycobacterium tuberculosis*. In that case, the person's ability to react to the TST antigen may wane over time and provide false-negative results. Therefore, our LTBI burden estimates may be an underestimate of the actual burden among HCWs in Bangladesh. Secondly, BCG vaccination might have affected TST positivity. A recent study showed that the possibility of false TST positivity might arise at a cut-off of TST>5mm as the result of BCG vaccine administered at infancy, TST indurations $\geq$ 10 mm appears unlikely to affect the diagnosis of LTBI [55]. Moreover, since all healthcare workers received BCG at birth, which is more than ten years prior to the TST, the result is less likely to be affected by the BCG exposures [56]. Thirdly, the study might have to overestimate TST positivity due to HCWs exposures to non-tuberculous mycobacteria, which are also prevalent in Bangladesh [11, 57]. However, in a systematic review, Farhat et al. (2006) showed that false-positive results due to non-tuberculous mycobacteria are not common in countries with high TB prevalence [56].

## Conclusions

In conclusion, this study identified a high number of HCWs was infected with LTBI in the study facilities. This study represents a first step in estimating the risk of TB exposures and LTBI among HCWs in public tertiary care hospitals in Bangladesh. Our study identified the importance of preventing or interrupting the occupational risk of TB among HCWs. Considering the high prevalence of LTBI among the HCWs, we recommend that NTP consider providing preventative therapy to the HCWs as the high-risk group. Our analysis also warrants the immediate implementation of TB IPC in Bangladeshi tertiary care teaching hospitals. Our study also recommends relocation or renovation of the nursing workstation to minimize exposure for staff. Separating the nursing station with transparent curtains or glass could be an option. We recommend establishing a TB IPC program in all tertiary care teaching hospitals to ensure proper implementation of the TB IPC healthcare measures. Finally, an assessment of the HCWs private chambers and private clinics should be done to recommend context-appropriate modifiable TB IPC measures, including improving the facilities' physical layout. The study findings will assist Bangladesh's national TB control program to effectively revise the TB IPC policy and reduce healthcare-associated TB infection.

## Supporting information

**S1 Dataset.**
(CSV)

**S1 File.**
(PDF)

## Acknowledgments

We would like to thank the study hospitals' directors and all the study participants for their time and respect.

The funders had no role in study design, data collection and analysis, decision to publish, or preparation of the manuscript.

A minimal dataset that supports the study has been attached as a supporting document. icddr,b's department of research administration maintains a data repository and a copy of the complete dataset will remain in the repository. Interested researchers may contact Ms. Armana Ahmed, head of research administration (aahmed@icddrb.org), for approval and full data access.

## Author Contributions

**Conceptualization:** Md Saiful Islam, Abrar Ahmad Chughtai, Muhammad Tauhidul Islam, Sayera Banu, Holly Seale.

**Data curation:** Md Saiful Islam, Arifa Nazneen, Kamal Ibne Amin Chowdhury, Muhammad Tauhidul Islam, Sayeeda Tarannum, S. M. Hasibul Islam.

**Formal analysis:** Md Saiful Islam, Abrar Ahmad Chughtai, Kamal Ibne Amin Chowdhury, Muhammad Tauhidul Islam, Sayeeda Tarannum, S. M. Hasibul Islam, Sayera Banu, Holly Seale.

**Funding acquisition:** Md Saiful Islam, Sayera Banu.

**Investigation:** Md Saiful Islam, Arifa Nazneen, Kamal Ibne Amin Chowdhury, Muhammad Tauhidul Islam, Sayeeda Tarannum, S. M. Hasibul Islam.

**Methodology:** Md Saiful Islam, Abrar Ahmad Chughtai, Kamal Ibne Amin Chowdhury, Muhammad Tauhidul Islam, Sayeeda Tarannum, S. M. Hasibul Islam, Sayera Banu, Holly Seale.

**Project administration:** Md Saiful Islam, Arifa Nazneen, Kamal Ibne Amin Chowdhury, Muhammad Tauhidul Islam, Sayeeda Tarannum, S. M. Hasibul Islam, Sayera Banu.

**Supervision:** Abrar Ahmad Chughtai, Sayera Banu, Holly Seale.

**Validation:** Abrar Ahmad Chughtai, Arifa Nazneen, Sayera Banu, Holly Seale.

**Writing – original draft:** Md Saiful Islam.

**Writing – review & editing:** Md Saiful Islam, Abrar Ahmad Chughtai, Arifa Nazneen, Kamal Ibne Amin Chowdhury, Muhammad Tauhidul Islam, Sayeeda Tarannum, S. M. Hasibul Islam, Sayera Banu, Holly Seale.

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
