## [Decision Letter · Decision Letter 0]

15 Oct 2020

PONE-D-20-30710

Are healthcare workers in public tertiary care general hospitals at risk of TB infection? A tuberculin skin test survey in two public tertiary care hospitals, Bangladesh

PLOS ONE

Dear Dr. Islam,

Thank you for submitting your manuscript to PLOS ONE. After careful consideration, we feel that it has merit but does not fully meet PLOS ONE’s publication criteria as it currently stands. Therefore, we invite you to submit a revised version of the manuscript that addresses the points raised during the review process.

We look forward to receiving your revised manuscript.

Kind regards,

Frederick Quinn

Academic Editor

PLOS ONE

Journal Requirements:

2. Regarding data availability, please note that PLOS journals require authors to make all data underlying the findings described in their manuscript fully available without restriction, with rare exception. PLOS requires that a “minimal data set” is shared, defined as the data set used to reach the conclusions drawn in the manuscript with related metadata and methods, and any additional data required to replicate the reported study findings in their entirety. Authors do not need to submit their entire data set if only a portion of the data were used in the reported study. Also, authors do not need to submit the raw data collected during an investigation if the standard in the field is to share data that have been processed. Please submit the following data: The values behind the means, standard deviations and other measures reported; The values used to build graphs; The points extracted from images for analysis.” Please review http://journals.plos.org/plosone/s/data-availability#loc-faqs-for-data-policy. If you are unable to share the data, this may result in manuscript rejection.

3. Please include additional information regarding the survey or questionnaire used in the study and ensure that you have provided sufficient details that others could replicate the analyses. For instance, if you developed a questionnaire as part of this study and it is not under a copyright more restrictive than CC-BY, please include a copy, in both the original language and English, as Supporting Information, or include a citation if it has been published previously.

4. In the Methods, please discuss whether and how the questionnaire was validated and/or pre-tested. If these did not occur, please provide the rationale for not doing so.

5. In your statistical analyses, please state whether you accounted for clustering by hospital. For example, did you consider using multilevel models?

6. Please revise overstated conclusions and statements that imply causal association. For example your statement that, "...poor implementation of TB IPC in public tertiary care teaching hospitals increased the risk of LTBI among HCWs...," implies causation. A cross-sectional study can not provide sufficient evidence for causal inferences should be avoided and instead indicate that associations were observed.

7.We note that you have indicated that data from this study are available upon request. PLOS only allows data to be available upon request if there are legal or ethical restrictions on sharing data publicly. For information on unacceptable data access restrictions, please see http://journals.plos.org/plosone/s/data-availability#loc-unacceptable-data-access-restrictions.

Reviewers' comments:

Reviewer's Responses to Questions

**Comments to the Author**

1. Is the manuscript technically sound, and do the data support the conclusions?

Reviewer #1: Yes

Reviewer #2: Yes

2. Has the statistical analysis been performed appropriately and rigorously? 

Reviewer #1: Yes

Reviewer #2: Yes

3. Have the authors made all data underlying the findings in their manuscript fully available?

Reviewer #1: No

Reviewer #2: No

4. Is the manuscript presented in an intelligible fashion and written in standard English?

Reviewer #1: Yes

Reviewer #2: Yes

5. Review Comments to the Author

Reviewer #1: What about the incidence of tuberculosis in Bangladesh?

What about the program of BCG vaccination in Bangladesh?

Did you use the QFT-TB gold test for diagnosis of TB infection or TB disease in Bangladesh? If yes, why you didn't use QFT and TST tests for this study of screening of risk to TB infection ?

Did you have the information of participants about different risk factors like asthma, diabetes or others factors ?

Could you explain the choice of the references of each parameter for calculation of odd ratio for tables 2 and 3?

Reviewer #2: 1. Title: a long one, no need for the question included,

a. Are healthcare workers in public tertiary care general hospitals at risk of TB infection?

b. all of us know the answer, so better remove it,

c. and only keep title starting A tuberculin skin test survey in two public tertiary care hospitals, Bangladesh

d. you cannot use; Prevalence of latent tuberculosis infection among healthcare workers,

Bangladesh, since its done in two specific hospitals that does not represent Bangladesh in general

2. ABSTRACT:

a. in multivariate analysis need to remove age and work years, because of problem of colinearity

b. there should be no discussion in abstract, you mean conclusion

c. second part of conclusion is not accepted because it is not based on study results.

3. Introduction is poor, need more elaboration on LTB as a major health problem for HCWs

4. Review of literature missing key and important publications in the context

5. In results 1st paragraph, 169 refused to participate, why, need explanation on that.

6. The last part of results are comments that have no tables, where are tables for use of masks and exposure times.

7. Where is the conclusion and recommendation?

6. PLOS authors have the option to publish the peer review history of their article (what does this mean?). If published, this will include your full peer review and any attached files.

Reviewer #1: No

Reviewer #2: **Yes: **Prof. Mostafa Abbas Kofi

---

## [Author Response · Author response to Decision Letter 0]

7 Nov 2020

PONE-D-20-30710

To

Frederick Quinn

Academic Editor

PLOS ONE

Dear Frederick Quinn:

We are thankful to the reviewers for their valuable feedback. We are also grateful to the editor for allowing us to respond to the comments. Based on the helpful feedback, we revised the manuscript and believe it is more precise, clear, and informative. The following is an itemized list of our specific responses to the reviewers' comments. We have highlighted the changes in the manuscript (marked) as well. 

As suggested, we have also revised the data availability statements. A minimal dataset that supports the study has been attached as a supporting document. icddr,b’s department of research administration maintains a data repository anda copy of the complete dataset will remain in the repository. Interested researchers may contact Ms. Armana Ahmed, head of research administration (aahmed@icddrb.org), for approval and full data access.

We would appreciate your further review. Please contact me directly with any additional questions or comments. We look forward to hearing from you.

Sincerely

Md. Saiful Islam

Corresponding Author

saiful@icddrb.org

Comments: When submitting your revision, we need you to address these additional requirements. Please ensure that your manuscript meets PLOS ONE's style requirements, including those for file naming. The PLOS ONE style templates can be found at

https://journals.plos.org/plosone/s/file?id=wjVg/PLOSOne_formatting_sample_main_body.pdf andhttps://journals.plos.org/plosone/s/file?id=ba62/PLOSOne_formatting_sample_title_authors_affiliations.pdf

Response: Thank you. The manuscript and the authors’ affiliation have been updated according to the PLOS ONE’s style requirements.

Comments: 2. Regarding data availability, please note that PLOS journals require authors to make all data underlying the findings described in their manuscript fully available without restriction, with rare exception. PLOS requires that a “minimal data set” is shared, defined as the data set used to reach the conclusions drawn in the manuscript with related metadata and methods, and any additional data required to replicate the reported study findings in their entirety. Authors do not need to submit their entire data set if only a portion of the data were used in the reported study. Also, authors do not need to submit the raw data collected during an investigation if the standard in the field is to share data that have been processed. Please submit the following data: The values behind the means, standard deviations and other measures reported; The values used to build graphs; The points extracted from images for analysis.” Please review http://journals.plos.org/plosone/s/data-availability#loc-faqs-for-data-policy. If you are unable to share the data, this may result in manuscript rejection.

Response: As recommended, we shared a minimal dataset.

Comment: 3. Please include additional information regarding the survey or questionnaire used in the study and ensure that you have provided sufficient details that others could replicate the analyses. For instance, if you developed a questionnaire as part of this study and it is not under a copyright more restrictive than CC-BY, please include a copy, in both the original language and English, as Supporting information, or include a citation if it has been published previously.

Response: As suggested, we have uploaded a copy of the questionnaire used in the survey.

Comment: 4. In the Methods, please discuss whether and how the questionnaire was validated and/or pre-tested. If these did not occur, please provide the rationale for not doing so.

Response: Thank you. We added, “The tool, adopted from a previous study, was updated and pre-tested for cognitive test among five HCWs in the study hospitals [1].” on page 6 in the manuscript (clean version).

Comment: 5. In your statistical analyses, please state whether you accounted for clustering by hospital. For example, did you consider using multilevel models?

 Response: Thanks for your comment. Based on your suggestion, we have revised the method and the analysis, and updated the table 2 and the text in the manuscript. Under the method section, we added, “In the multivariate analysis, we used a generalized linear model adjusting for hospital level clustering effect (using ‘glm.cluster’ command from the miceadds R package)” on page 7.

Comments: 6. Please revise overstated conclusions and statements that imply causal association. For example your statement that, "...poor implementation of TB IPC in public tertiary care teaching hospitals increased the risk of LTBI among HCWs...," implies causation. A cross-sectional study cannot provide sufficient evidence for causal inferences should be avoided and instead indicate that associations were observed.

 Response: Based on your suggestion, we have revised the conclusion as, “This study identified high prevalence of LTBI among HCWs. This may be due to the level of exposure to pulmonary TB patients, and/or limited use of personal protective equipment along with poor implementation of TB IPC in the hospitals. Considering the high prevalence of LTBI, we recommend the national TB program consider providing preventative therapy to the HCWs as the high-risk group, and implement TB IPC in the hospitals” on page 2.

Comments: 7.We note that you have indicated that data from this study are available upon request. PLOS only allows data to be available upon request if there are legal or ethical restrictions on sharing data publicly. For information on unacceptable data access restrictions, please see http://journals.plos.org/plosone/s/data-availability#loc-unacceptable-data-access-restrictions.

Response: Now, we have revised the statements. A minimal dataset that supports the study has been attached as a supporting document.

Review Comments to the Author

Reviewer #1: 

Comment: What about the incidence of tuberculosis in Bangladesh?

Response: We have added this information, “Bangladesh shares 3.6% of the global total of 10 million people estimated with TB diseases in 2019[2]. Bangladesh is one of the 22 high TB burden countries in the world with an estimated incidence for all forms of TB in 2019 was 221 (uncertainty interval: 161-291) per 100 000 population[2] on page 4 (Manuscript clean version).

Comment: What about the program of BCG vaccination in Bangladesh?

Response: Under the introduction, we have added this information, “In line with the country's TB epidemiology, Bangladesh's extended program on immunization has included neonatal bacillus Calmette-Guérin (BCG) vaccination nationwide since the 1980s [3]. The national coverage of the BCG vaccine was 86% in 1991; 95% in 2000; and 99% since 2013[3]” on page 4.

Comments: Did you use the QFT-TB gold test for diagnosis of TB infection or TB disease in Bangladesh? If yes, why you didn't use QFT and TST tests for this study of screening of risk to TB infection?

Response: Thank you for your valuable comments. QuantiFerron-Gold-in-Tube test (QFT-GIT) is expensive and is not routinely use in Bangladesh. Moreover, QFT-GIT requires a specialized laboratory for sample processing that was not widely available outside Dhaka during the time we conducted this study. Therefore, we could not use the QFT-GIT test.

Comment: Did you have the information of participants about different risk factors like asthma, diabetes or others factors?

Response: No, we did not collect data on asthma and diabetes. However we collected information on smoking history. Please see page 8 and the table 1 and table 2. 

Comments: Could you explain the choice of the references of each parameter for calculation of odd ratio for tables 2 and 3?

Response: The reference category was chosen either with the highest frequency or the group with a lower risk of infection based on published literature.

Reviewer #2: 

Comment: 1. Title: a long one, no need for the question included,

a. Are healthcare workers in public tertiary care general hospitals at risk of TB infection?

Response: Thank you. We have revised the title as, “A tuberculin skin test survey among healthcare workers in two public tertiary care hospitals in Bangladesh."

Comment: b. all of us know the answer, so better remove it,

Response: As suggested, we have removed “Are healthcare workers in public tertiary care general hospitals at risk of TB infection?” from the title.

Comment: c. and only keep title starting A tuberculin skin test survey in two public tertiary care hospitals, Bangladesh

Response: Thank you. We have revised the title as, ““A tuberculin skin test survey among healthcare workers in two public tertiary care hospitals in Bangladesh."

Comment: d. you cannot use; Prevalence of latent tuberculosis infection among healthcare workers, Bangladesh, since its done in two specific hospitals that does not represent Bangladesh in general

Response: We revised it as, “Prevalence of latent tuberculosis infection among healthcare workers in two public tertiary care hospitals in Bangladesh”.

Comment: 2. ABSTRACT:

a. in multivariate analysis need to remove age and work years, because of problem of colinearity

Response: Thank you. On page 7, we mentioned, “To check multicollinearity, we performed a variance inflation factor (VIF) analysis, and factors with VIF> 5, we considered them collinear and excluded from the multivariate model [4]. 

We checked for multicollinearity, and the variables with <5 variance inflation factors have been added in the multivariate analysis. Please see the outcome of the multicollinearity test.

Comment: b. there should be no discussion in abstract, you mean conclusion

Response: Thank you. We have revised it. 

Comment: c. second part of conclusion is not accepted because it is not based on study results.

Response: We have revised the concluding sentence. The revised sentence is, “This study identified high prevalence of LTBI among HCWs. This may be due to the level of exposure to pulmonary TB patients, and/or limited use of personal protective equipment along with poor implementation of TB IPC in the hospitals. Considering the high prevalence of LTBI, we recommend the national TB program consider providing preventative therapy to the HCWs as the high-risk group, and implement TB IPC in the hospitals on pages 2 (manuscript clean version).

Comment: 3. Introduction is poor, need more elaboration on LTB as a major health problem for HCWs

Response: Thank you. We have now added more information to strengthen the introduction. On page 3 we added more information and revised it as, “In 2019, the WHO estimated that 10 million people developed TB disease globally, of whom 44% were from South-East Asia [2]. Moreover, 22,314 healthcare workers (HCWs) developed TB in the same year, with most coming from high TB burden countries in Asia [2]. Hospital HCWs in high TB burden countries are at increased risk of TB infection due to their exposure to a higher number of pulmonary TB patients than the hospital HCWs working in low TB-incidence countries [5]. A recent mathematical modelling study on the global burden of latent TB infection (LTBI) estimated that around 1.7 billion people are infected with LTBI in the world [6]. In a systematic review of 18 studies from seven high TB burden countries, the prevalence of LTBI among HCWs was reported to be 47% (95% CI 34- 60)[7]. This risk may be high among HCWs who work in health facilities that lack proper infrastructure and limited implementation of TB infection prevention and control (IPC) healthcare measures. The risk of LTBI among different health care workers may vary by place of work, duration of exposures, and compliance with TB IPC measures [8]. Prior studies identified considerable heterogeneity in the risk of LTBI among different occupations and reported high risk among doctors, nurses, and ancillary workers [8, 9]. HCWs that were most likely to be infected had the most prolonged duration and extent of patient contact [9, 10]. 

People with LTBI represent a reservoir for potential TB disease [6]. Overall, without treatment, about 5 to 10% of infected persons may develop TB disease at some time in their lives, and the active stage frequently occurs within the first two years after infection [11, 12]. People with an impaired immune system are at increased risk of developing TB disease than persons with standard immune systems[12]. Diabetes and smoking behavior also increase TB disease risk among the person with LTBI[13].

On page 4, we added, “Bangladesh shares 3.6% of the global total of 10 million people estimated with TB diseases in 2019[2]. Bangladesh is one of the 22 high TB burden countries in the world with an estimated incidence for all forms of TB in 2019 was 221 (uncertainty interval: 161-291) per 100 000 population[2]. In line with the country's TB epidemiology, Bangladesh's extended program on immunization has included neonatal bacillus Calmette-Guérin (BCG) vaccination nationwide since the 1980s [3]. The national coverage of the BCG vaccine was 86% in 1991; 95% in 2000; and 99% since 2013[3].”

Comment: 4. Review of literature missing key and important publications in the context

Response: Thank you. I would like to request you to see our response under the previous comment. We have updated literature review and added recent publications. 

Comment: 5. In results 1st paragraph, 169 refused to participate, why, need explanation on that.

Response: On page 10, we added, “One hundred and fifty nine HCWs did not consent for TST due to their illness, prior history of allergic reaction to TST, unavailability for TST reading, and past history of active TB disease”.

Comments: 6. The last part of results are comments that have no tables, where are tables for use of masks and exposure times.

Response: We have now added the information in table 2.

Demography and exposures TST positive % (n/N) TST negative % (n/N) OR*(95% CI*) aOR*(95% CI*)

Hours working per day

<6 hours 43 (15/35) 57(20/35) Reference 

6 hours and above 43(303/704) 57 (401/704) 1.01 (0.51-2.03) 

Use of mask or respirator

Yes 40 (75/190) 60 (115/190) 0.85(0.61-1.19) 

No 43 (271/626) 57 (355/626) Reference 

Comments: 7. Where are the conclusion and recommendation?

Response: On Page 13 and 14, we described the conclusion and recommendations, " In conclusion, this study identified a high number of HCWs was infected with LTBI in the study facilities. This study represents a first step in estimating the risk of TB exposures and LTBI among HCWs in public tertiary care hospitals in Bangladesh. Our study identified the importance of preventing or interrupting the occupational risk of TB among HCWs. Considering the high prevalence of LTBI among the HCWs, we recommend that NTP consider providing preventative therapy to the HCWs as the high-risk group. Our analysis also warrants the immediate implementation of TB IPC in Bangladeshi tertiary care teaching hospitals. Our study also recommends relocation or renovation of the nursing workstation to minimize exposure for staff. Separating the nursing station with transparent curtains or glass could be an option. We recommend establishing a TB IPC program in all tertiary care teaching hospitals to ensure proper implementation of the TB IPC healthcare measures. Finally, an assessment of the HCWs private chambers and private clinics should be done to recommend context-appropriate modifiable TB IPC measures, including improving the facilities' physical layout. The study findings will assist Bangladesh’s national TB control program to effectively revise the TB IPC policy and reduce healthcare-associated TB infection.”

References:

1. Islam MS. Latent tuberculosis infection among healthcare workers in chest disease hospitals, Bangladesh. Health and Science Bulletin. 2014;12(1):1-7.

2. World Health Organization. Global Tuberculosis Report. Geneva: World Health Organization, 2020.

3. Sarkar PK, Sarker NK, Doulah S, Bari TI. Expanded Programme on Immunization in Bangladesh: A Success Story. Bangladesh Journal Of Child Health. 2015;39(2):93-8.

4. Kwon YS, Kim YH, Jeon K, Jeong BH, Ryu YJ, Choi JC, et al. Factors that Predict Negative Results of QuantiFERON-TB Gold In-Tube Test in Patients with Culture-Confirmed Tuberculosis: A Multicenter Retrospective Cohort Study. PloS one. 2015;10(6):e0129792. Epub 2015/06/13. doi: 10.1371/journal.pone.0129792. PubMed PMID: 26070207; PubMed Central PMCID: PMCPMC4466377.

5. Uden L, Barber E, Ford N, Cooke GS. Risk of Tuberculosis Infection and Disease for Health Care Workers: An Updated Meta-Analysis. Open forum infectious diseases. 2017;4(3):ofx137. Epub 2017/09/07. doi: 10.1093/ofid/ofx137. PubMed PMID: 28875155; PubMed Central PMCID: PMCPMC5575844.

6. Houben RM, Dodd PJ. The Global Burden of Latent Tuberculosis Infection: A Re-estimation Using Mathematical Modelling. PLoS medicine. 2016;13(10):e1002152. Epub 2016/10/26. doi: 10.1371/journal.pmed.1002152. PubMed PMID: 27780211; PubMed Central PMCID: PMCPMC5079585.

7. Nasreen S, Shokoohi M, Malvankar-Mehta MS. Prevalence of Latent Tuberculosis among Health Care Workers in High Burden Countries: A Systematic Review and Meta-Analysis. PloS one. 2016;11(10):e0164034. Epub 2016/10/07. doi: 10.1371/journal.pone.0164034. PubMed PMID: 27711155; PubMed Central PMCID: PMCPMC5053544.

8. Joshi R, Reingold AL, Menzies D, Pai M. Tuberculosis among health-care workers in low- and middle-income countries: a systematic review. PLoS medicine. 2006;3(12):e494. doi: 10.1371/journal.pmed.0030494. PubMed PMID: 17194191; PubMed Central PMCID: PMC1716189.

9. Gopinath K, Siddique S, Kirubakaran H, Shanmugam A, Mathai E, Chandy G. Tuberculosis among healthcare workers in a tertiary-care hospital in South India. Journal of Hospital Infection. 2004;57(4):339-42.

10. Donald PR, Helden PDv. The Global Burden of Tuberculosis — Combating Drug Resistance in Difficult Times. New England Journal of Medicine. 2009;360(23):2393-5.

11. Lillebaek T, Dirksen A, Baess I, Strunge B, Thomsen VØ, Andersen ÅB. Molecular evidence of endogenous reactivation of Mycobacterium tuberculosis after 33 years of latent infection. The Journal of infectious diseases. 2002;185(3):401-4.

12. Centers for Disease Control and Prevention. TB Risk Factors Atlanta, Georgia, USA: Centers for Disease Control and Prevention; 2016 [cited 2020 26 October]. Available from: https://www.cdc.gov/tb/topic/basics/risk.htm.

13. Gajalakshmi V, Peto R, Kanaka TS, Jha P. Smoking and mortality from tuberculosis and other diseases in India: retrospective study of 43 000 adult male deaths and 35 000 controls. The Lancet. 2003;362(9383):507-15.

---

## [Decision Letter · Decision Letter 1]

1 Dec 2020

A tuberculin skin test survey among healthcare workers in two public tertiary care hospitals in Bangladesh

PONE-D-20-30710R1

Dear Dr. Islam,

We’re pleased to inform you that your manuscript has been judged scientifically suitable for publication and will be formally accepted for publication once it meets all outstanding technical requirements.

Kind regards,

Frederick Quinn

Academic Editor

PLOS ONE

Additional Editor Comments (optional):

Reviewers' comments:

Reviewer's Responses to Questions

**Comments to the Author**

1. If the authors have adequately addressed your comments raised in a previous round of review and you feel that this manuscript is now acceptable for publication, you may indicate that here to bypass the “Comments to the Author” section, enter your conflict of interest statement in the “Confidential to Editor” section, and submit your "Accept" recommendation.

Reviewer #1: All comments have been addressed

Reviewer #2: All comments have been addressed

2. Is the manuscript technically sound, and do the data support the conclusions?

Reviewer #1: Yes

Reviewer #2: Yes

3. Has the statistical analysis been performed appropriately and rigorously? 

Reviewer #1: Yes

Reviewer #2: Yes

4. Have the authors made all data underlying the findings in their manuscript fully available?

Reviewer #1: Yes

Reviewer #2: Yes

5. Is the manuscript presented in an intelligible fashion and written in standard English?

Reviewer #1: Yes

Reviewer #2: Yes

6. Review Comments to the Author

Reviewer #1: (No Response)

Reviewer #2: questions were addressed by authors.

its important to hare your publication on research website such a research gate and LinkedIn.

7. PLOS authors have the option to publish the peer review history of their article (what does this mean?). If published, this will include your full peer review and any attached files.

Reviewer #1: No

Reviewer #2: No

---

## [Editor Report · Acceptance letter]

4 Dec 2020

PONE-D-20-30710R1 

A tuberculin skin test survey among healthcare workers in two public tertiary care hospitals in Bangladesh 

Dear Dr. Islam:

I'm pleased to inform you that your manuscript has been deemed suitable for publication in PLOS ONE. Congratulations! Your manuscript is now with our production department. 

Kind regards, 

on behalf of

Dr. Frederick Quinn 

Academic Editor

PLOS ONE